

# DSCOVR/EPIC-derived global hourly/daily downward shortwave and photosynthetically active radiation data at 0.1°×0.1° resolution

Dalei Hao[1,2,3], Ghassem R. Asrar[4], Yelu Zeng[1], Qing Zhu[5], Jianguang Wen[2,3], Qing Xiao[2,3], Min Chen[1]

[1]Joint Global Change Research Institute, Pacific Northwest National Laboratory, College Park, MD 20740, USA
[2]State Key Laboratory of Remote Sensing Science, Aerospace Information Research Institute, Chinese Academy of Sciences, Beijing 100101, China
[3]University of Chinese Academy of Sciences, Beijing 100049, China
[4]Universities Space Research Association, Columbia, MD 21046, USA
[5]Earth Science Division, Lawrence Berkeley National Lab, Berkeley, CA 94720, USA

*Correspondence to*: Min Chen (min.chen@pnnl.gov)

**Abstract.** Downward shortwave radiation (SW) and photosynthetically active radiation (PAR) play crucial roles in Earth system dynamics. Spaceborne remote sensing techniques provide a unique means for mapping accurate spatio-temporally-continuous SW/PAR, globally. However, any individual polar-orbiting or geostationary satellite cannot satisfy the desired high temporal

resolution (sub-daily) and global coverage simultaneously, while integrating and fusing multi-source data from complementary satellites/sensors is challenging because of co-registration, inter-calibration, near real-time data delivery and the effects of discrepancies in orbital geometry. The Earth Polychromatic Imaging Camera (EPIC) onboard the Deep Space Climate Observatory (DSCOVR), launched in February 2015, offers an unprecedented possibility to bridge the gap between high temporal resolution and global coverage, and characterize the diurnal cycles of SW/PAR globally. In this study, we adopted a suite of well-validated

data-driven machine-learning models to generate the first global land products of SW/PAR, from June 2015 to June 2019, based on DSCOVR/EPIC data. The derived products have high temporal resolution (hourly) and medium spatial resolution (0.1°×0.1°), and include estimates of the direct and diffuse components of SW/PAR. We used independently widely-distributed ground station data from the Baseline Surface Radiation Network (BSRN), the Surface Radiation Budget Network (SURFRAD), NOAA's Global Monitoring Division and the U.S. Department of Energy's Atmospheric System Research (ASR) program to evaluate the

performance of our products, and further analyzed and compared the spatio-temporal characteristics of the derived products with the benchmarking Clouds and the Earth's Radiant Energy System Synoptic (CERES) data. We found both the hourly and daily products to be consistent with ground-based observations (e.g., hourly and daily total SWs have low biases of -3.96 and -0.71 W/m$^2$ and root mean square errors (RMSEs) of 103.50 and 35.40 W/m$^2$, respectively). The developed products capture the complex spatio-temporal patterns well and accurately track substantial diurnal, monthly, and seasonal variations of SW/PAR when

compared to CERES data. They provide a reliable and valuable alternative for solar photovoltaic applications worldwide and can be used to improve our understanding of the diurnal and seasonal variabilities of the terrestrial water, carbon and energy fluxes at various spatial scales. The products are freely available at https://doi.org/10.25584/1595069 (Hao et al., 2020).

## 1 Introduction

Downward shortwave radiation (SW) and photosynthetically active radiation (PAR) profoundly affect the terrestrial environment
(Wild et al., 2005), and are fundamental for global energy balance (Liang et al., 2010), carbon budget (Farquhar and Roderick, 2003), hydrological cycle (Roderick and Farquhar, 2002), and solar energy production and utilization (Sweerts et al., 2019). Partitioning total SW/PAR into their direct and diffuse components also is important for solar resource management and



photovoltaic power design (Khahro et al., 2015;Raptis et al., 2017), and terrestrial photosynthesis estimations (Mercado et al., 2009;Gu et al., 2002;Chen and Zhuang, 2014;Wang et al., 2018).

Satellite remote sensing has been widely used to map SW/PAR across various spatial and temporal scales (Pinker et al., 2005;Huang et al., 2019). Traditional ground-based observations have the required high accuracy but sparse point-specific distributions, and thus inadequate spatial representation (Korany et al., 2016), while numerical modelling with spatio-temporally-continuous mapping has relatively low spatial resolution and large errors and uncertainties (Zhao et al., 2013). In contrast, remote sensing offers a more reliable and efficient tool to estimate high-quality SW/PAR globally with high spatio-temporal resolution,

as it characterizes heterogeneous spatial distributions and captures the complex dynamic evolution of atmosphere, cloud and land surface processes at regional and even global scales (Huang et al., 2019;Li et al., 2020). Currently, a series of remote sensing-based SW/PAR datasets/products are available: 1) from polar-orbiting satellites, e.g., Breathing Earth System Simulator (BESS) (Ryu et al., 2018) and MODIS MCD18 products (Wang et al., 2020); 2) from geostationary satellites, e.g., Himawari-8/Advanced Himawari Imager (AHI) (Letu et al., 2020); and 3) from fusing multi-source data/observations, such as the Global LAnd Surface

Satellite (GLASS) (Zhang et al., 2014) and the Clouds and the Earth's Radiant Energy System Synoptic (CERES) (Rutan et al., 2015).

Global high-quality SW/PAR data at sub-daily scales are highly desired for investigating the diurnal variabilities of solar-induced fluorescence, photosynthesis (Damm et al., 2010) and evapotranspiration (Van Heerwaarden et al., 2010), and for solar energy utilization (Sweerts et al., 2019). However, accurately quantifying global SW/PAR is challenging based on current polar-

orbiting or/and geostationary satellites/sensors, because: 1) sun-synchronous polar-orbiting satellites generally have high spatial resolution but cannot capture the sub-daily variations of SW/PAR owing to low revisiting frequency; 2) geostationary satellites usually have high temporal resolution but limited geographical coverage (i.e. several different satellites systems for covering the entire Earth); 3) fusing multi-source data acquired from complementary satellites/sensors is challenging due to the issues of co-registration, inter-calibration, the effects of different orbital geometries and the difficulty of processing and delivering the finial

products in near-real time to users. In addition, most of the current remotely sensed SW/PAR estimations are conducted under the assumption of an independent pixel approximation (IPA) and simply neglect the three-dimensional (3D) radiative effects caused by inhomogeneous cloud fields (Wyser et al., 2005). The 3D effects can significantly influence the accuracy and quality of high-temporal-resolution SW/PAR estimations and are perhaps the largest error source for SW/PAR retrievals (Wyser et al., 2005;Huang et al., 2019). Although several methods have been developed based on full 3D radiative transfer models (Liou et al., 2013), there

is currently no practical and computationally-feasible approach to eliminate 3D radiative effects efficiently and completely (Huang et al., 2019).

The Deep Space Climate Observatory (DSCOVR), launched on February, 2015, leads a new era of monitoring the sun and Earth from deep space around the sun-Earth first Lagrange (L1) point (Burt and Smith, 2012). Its advanced Earth-facing camera, Earth Polychromatic Imaging Camera (EPIC), onboard DSCOVR, views nearly the entire sunlit part of the Earth, from pole to

pole, in near backscattering directions with 10 spectral bands from the ultra-violet to near-infrared wavelengths every 1~2 hours, giving EPIC a unique capability of monitoring and capturing the diurnal variation of ozone, clouds, aerosols, and vegetation properties (Marshak et al., 2018). DSCOVR/EPIC thus provides an unrivalled tool to capture the diurnal cycles of SW/PAR globally and overcomes some limitations of current remote sensing-based SW/PAR estimations. Compared to any individual polar-orbiting and geostationary satellite, DSCOVR/EPIC essentially bridges the gap between high revisiting frequency and global

coverage. Compared to the multi-source integration, the single DSCOVR/EPIC instrument avoids the compatibility and matching issues of using different sensors/satellites, and it is more suitable for processing and delivering the final products in real-time or near real-time to users. Fortunately, DSCOVR/EPIC is also characterized by a nearly constant scattering phase angle (angle formed



between the incident and scattered-to-satellite sunlight vectors) from 168.5° to 175.5°, which implies that DSCOVR/EPIC guarantees that the atmospheric column determining SW/PAR is nearly the same as that observed by the satellite. Therefore,

DSCOVR/EPIC has the potential to reduce significantly or eliminate completely the 3D radiative effects on the final products.

The overarching goal of this study is to 1) develop, document and present DSCOVR/EPIC-derived SW/PAR products covering a period of about 4 years (from June, 2015 to June, 2019) based on a suite of well-validated machine learning methods (Hao et al., 2019) and 2) perform a systematic and comprehensive assessment of the accuracy, consistency and spatio-temporal patterns of these products against comparable but independently developed and published data/products; and 3) make the resulting

dataset openly available for use by Earth system research and modelling, and for solar energy productions and use. The newly generated products are the first available SW/PAR products with high temporal frequency (hourly) and global coverage at a spatial resolution of 0.1°×0.1°, where the aggregated daily-scale data are available, and the direct and diffuse components of SW/PAR are also provided. We evaluate them against widely-distributed ground station data, analyze their spatio-temporal variations, and compared them to the widely-used CERES products. Finally, possible sources of uncertainties and potential improvements in the

future study are discussed.

## 2 Material and methods

### 2.1 Remote sensing data

The DSCOVR/EPIC science team has routinely developed and published a suite of official Level 2 (L2) products from DSCOVR/EPIC observations (Marshak et al., 2018), including stratospheric ozone concentrations (Herman et al., 2018), sulfur

dioxide ($SO_2$) from volcanic eruptions, atmospheric aerosols in the UV and visible spectral ranges, cloud parameters (Yang et al., 2019), atmospherically corrected land-surface reflectance and vegetation properties (Yang et al., 2017). These standard EPIC L2 products are publicly available from the NASA Langley Atmospheric Science Data Center and described in detail at https://eosweb.larc.nasa.gov/project/dscovr/dscovr_table. For this study, we obtained solar zenith angle, surface pressure, aerosol optical depth, cloud fraction from L2 aerosol product, cloud optical thickness and the most likely cloud phase from L2 cloud

products, and total column ozone from L2 ozone product, as well as the available quality flags for these products. We re-projected all datasets into global latitude/longitude grids with a spatial resolution of 0.1°×0.1° using the nearest neighbourhood resampling method.

As a key component of the Earth Observing System (EOS) program, the CERES project has developed and published globally long-term observed top-of-the-atmosphere (TOA) and calculated surface fluxes for study of climate and cloud

feedback (Wielicki et al., 1996). CERES Synoptic 1° (SYN1deg) Edition 4.1 products, released on August 22, 2019, contain global 1°×1° gridded monthly, monthly hourly, daily, 3-hourly and hourly averaged TOA and surface fluxes (Rutan et al., 2015). In particular, the SYN1deg Edition 4.1 products provide diurnally complete SW/PAR and their direct and diffuse components. However, the SYN1deg Edition 4.1 products are not suitable for inferring long-term trends of surface fluxes, due to limited climate quality. The CERES Energy Balanced and Filled (EBAF) Edition 4.1 products, released on May 28, 2019, provide global 1°×1°

gridded monthly averaged TOA and surface fluxes (Loeb et al., 2018;Kato et al., 2018). The CERES EBAF products are designed specifically for climate model evaluation and energy budget estimation, and are more suitable for long-term analysis of variability of SW/PAR (e.g. intra-seasonal and inter-annual changes). Both the SYN1deg and EBAF products are freely accessible via the CERES Visualization, Ordering and Subsetting Tool (https://ceres.larc.nasa.gov/order_data.php). In this study, we used both hourly and daily CERES SYN1deg Edition 4.1 products as a reference to evaluate the spatio-temporal patterns of EPIC-derived



SW/PAR products at both hourly and daily scales, and used the CERES EBAF Edition 4.1 products as a benchmark to evaluate the monthly and seasonal variations of EPIC-derived SW/PAR products.

## 2.2 Ground-based observation data

Ground-based measurements with high-quality instrumentation and long-term maintenance provide the most reliable and accurate SW/PAR data, which are generally deemed as the ground truth for evaluating the performance of remote sensing products. Since

1992, the international Baseline Surface Radiation Network (BSRN) under the World Climate Research Programme (WCRP) has provided high quality, high temporal resolution (1 min) ground-based radiation measurements of direct, diffuse and total SWs (Driemel et al., 2018;Ohmura et al., 1998). The BSRN stations are placed strategically to be representative of their relatively large surrounding area, thus suitable for the evaluation of satellite data. The Surface Radiation Budget Network (SURFRAD) supported by the NOAA Climate Program Office is operating in climatologically diverse regions and measuring accurate, continuous, long-

term surface radiation budget, and meteorological parameters routinely to support climate and weather studies over the United States (Augustine et al., 2000). The SURFRAD sites also measure and provide PAR data. The CERES/ARM Validation Experiment (CAVE) collected 58 land surface sites from the BSRN, NOAA's Global Monitoring Division, SURFRAD, and the U.S. Department of Energy's Atmospheric System Research (ASR) program and some personal communications (Rutan et al., 2001;Rutan et al., 2015). In the CAVE, all original 1 min data were averaged to different temporal scales (i.e. hourly, daily and

monthly) through the strict quality control and gap-filling using the linear interpolation. The CAVE dataset provides SW and its direct and diffuse components, but does not include PAR measurements. Further information on CAVE can be found at http://www-cave.larc.nasa.gov/.

We used the CAVE datasets to evaluate the performance of EPIC-derived SW products, and used the SURFRAD datasets to evaluate the performance of EPIC-derived PAR products. The original SURFRAD data were first gap-filled using a linear

interpolation technique and then temporally aggregated to both hourly and daily scales. Considering that some data from June, 2015 to December, 2016 were used to train and test the machine learning models (Hao et al., 2019), we used only all available data from 43 CAVE and 7 SURFARAD sites from January, 2017 to June, 2019. **Fig. 1** shows the geographical distribution of ground-based observation stations for evaluation in the study. These sites are further classified into three groups of polar (Arctic or Antarctic), island or coastal, and continental sites.


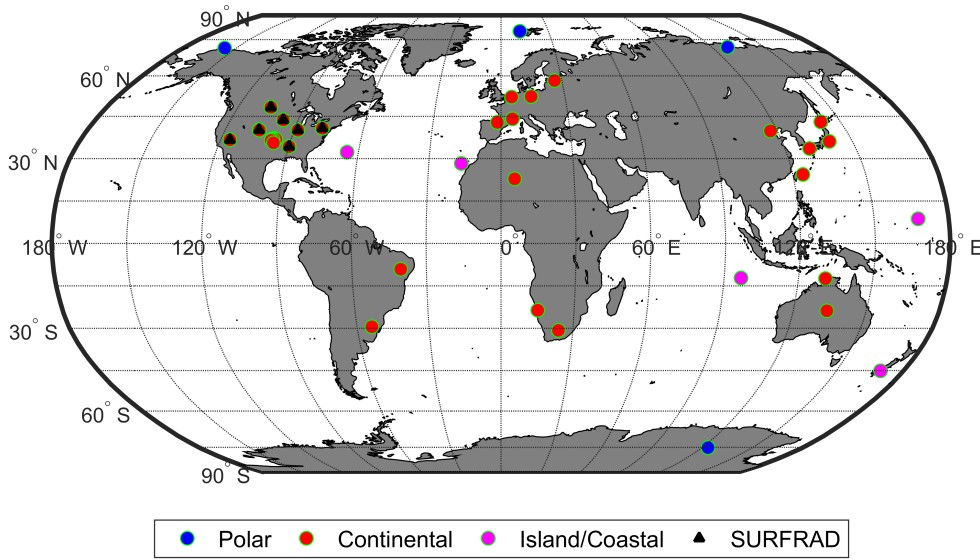

**Figure 1.** Geographical distribution of *in situ* observation sites for used for evaluation of space-based products. The blue, red and magenta circles (with green border) denote the polar (Arctic or Antarctic), island or coastal, and continental sites, respectively; and the black triangle represents the SURFRAD sites.

### 2.3 Estimation of SW/PAR fluxes

We adopted the trained random forest models developed by Hao et al. (2019) to estimate global SW/PAR from DSCOVR/EPIC datasets. The evaluation conducted by (Hao et al., 2019) showed that the random forest models perform very well against the ground measurements from BRSN and SURFRAD sites. In the study, we acquired hourly driving data for estimating SW/PAR based on the collected DSCOVR/EPIC L2 products (introduced in **Section 2.1**). We used the nearest neighbour interpolation approach to fill any gaps, based on the assumption that the atmospheric and cloud conditions remain unchanged and only solar zenith angle varies during a short period (1~2 hours). Hourly SW/PAR estimates were then produced using the random forest models and were aggregated into daily products. However, we found that gaps existed in the derived SW/PAR products due to failed retrievals of atmospheric and cloud parameters, especially in Arctic and Antarctic polar regions. Therefore, we used the CERES SYN1deg Edition 4.1 products to fill these gaps, based on linear interpolation techniques. We assigned quality flags to the derived products to denote: 0: successfully estimated from DSCOVR/EPIC; 1: gap-filled based on CERES data; 2: missing data.

## 3 Results

### 3.1 Evaluation of estimated SW and PAR against ground-based observations

#### 3.1.1 Overall performance of derived products

The hourly EPIC-derived diffuse, direct and total SW/PAR products match very well overall with the ground-based observations **(Fig. 2)**. For diffuse SW, the bias, root mean square error (RMSE), relative RMSE (RMSE to mean value, RRMSE) and the coefficient of determination ($R^2$) are 9.8 W/m², 74.97 W/m², 55.21% and 0.60, respectively. Direct SW has negative bias of -16.39 W/m², relatively large RMSE of 137.24 W/m² and RRMSE of 56.17%, and $R^2$ of 0.73. By contrast, both total SW and PAR have better performance with low biases (-3.96 and 7.31 W/m²), smaller RMSEs (103.50 and 50.44 W/m²) and RRMSEs (28.40% and





32.49%), and high $R^2$ values (0.87 and 0.83). These statistical metrics indicate that EPIC-based hourly SW/PAR estimates are

comparable to or better than other remote sensing-based products, e.g., Himawari-8/AHI-derived total SW has similar RMSE of 101.86 W/m$^2$ and $R^2$ of 0.87 (Letu et al., 2020).

The daily SW/PAR estimates are well correlated with the ground-based observations (**Fig. 3)**. Diffuse SW has a positive bias (5.25 W/m$^2$), a relatively small RMSE (25.25 W/m$^2$) but a large RRMSE (37.12%), and an $R^2$ of 0.65. By contrast, Direct SW has a negative bias (-6.09 W/m$^2$), a relatively large RMSE (45.46 W/m$^2$), a large RRMSE (39.49%) and a $R^2$ of 0.77. Total SW shows

good performance with a low bias of -0.71 W/m$^2$, a RMSE of 35.40 W/m$^2$, a smaller RRMSE of 19.45%, and high $R^2$ of 0.87. Total PAR also shows good relationship with the ground-based data (positive bias of 4.08 W/m$^2$, small RMSE of 16.80 W/m$^2$ and RRMSE of 21.88% high $R^2$ of 0.85). These results indicate that our daily products show comparable or better performance compared to other SW/PAR products, e.g., for total SW, MCD18 and GLASS products have similar RMSEs of 32.3 and 35.9 W/m$^2$ and higher biases of -7.8 and -7.6 W/m$^2$ (Wang et al., 2020).


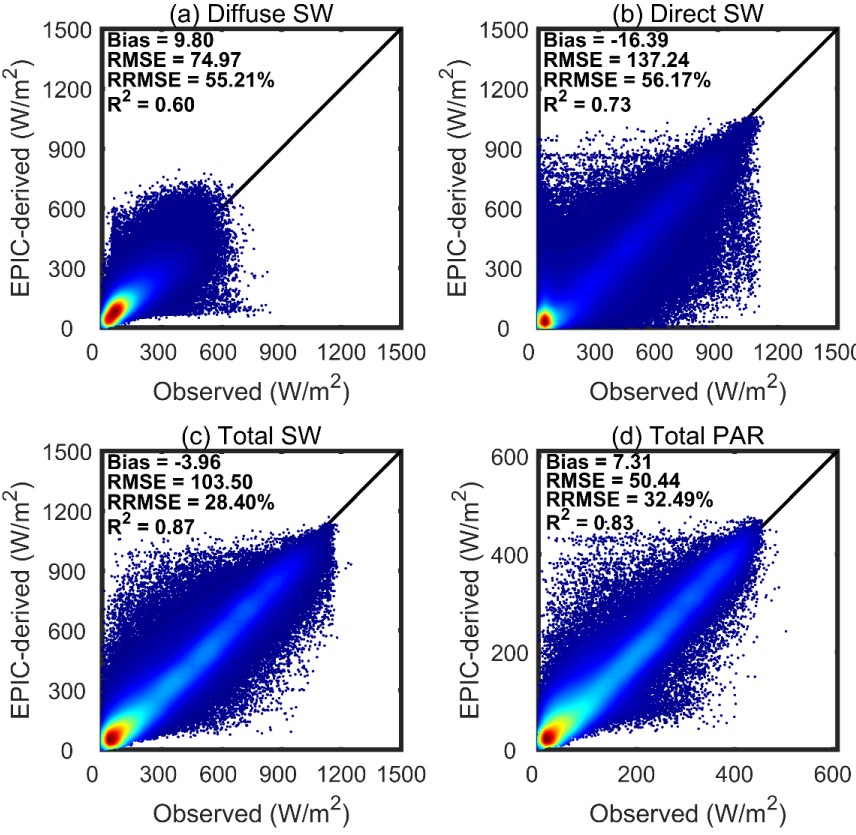

**Figure 2**. Evaluation of EPIC-based hourly SW/PAR estimates against ground-based observations.


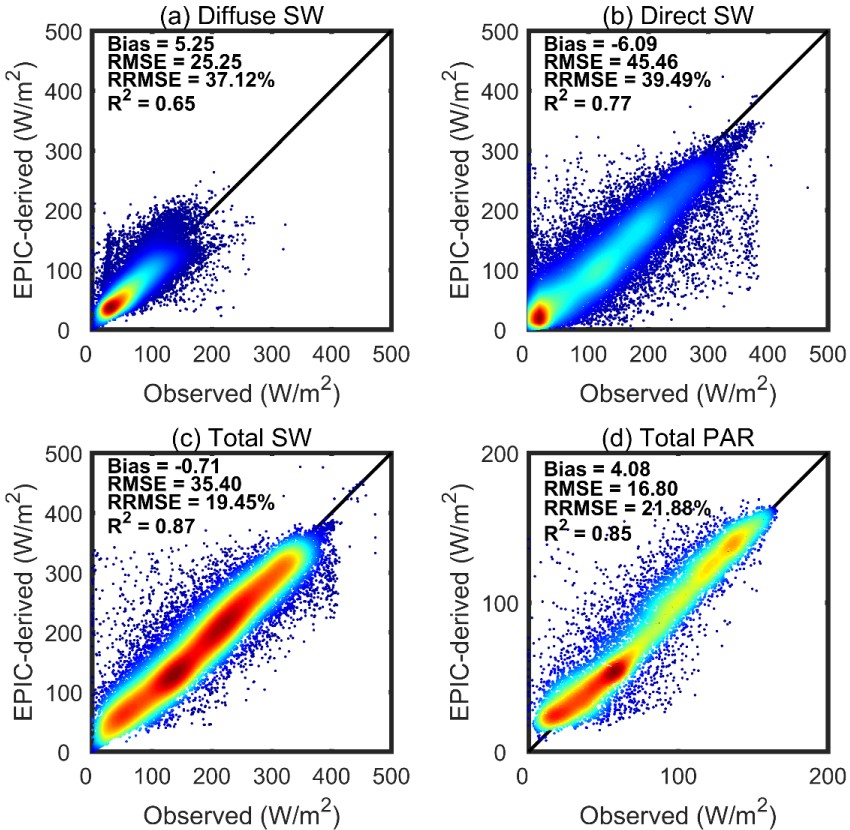

**Figure 3**. Evaluation of EPIC-based daily SW/PAR estimates against ground-based observations.

### 3.1.2 Temporal distribution of estimation errors of derived products

We first distinguish different sky conditions based on the ratio of diffuse to total SW (skyl): 1) clear: skyl < 0.3 during 70% time of one day; 2) overcast: skyl > 0.7 during 70% time of one day); and 3) cloudy: all the other cases. **Fig. 4** shows the comparisons of diurnal variations of both EPIC- and ground-based observed total SWs, averaged, during June, 2015-June, 2019 at 7 SURFRAD

sites under different sky conditions. For clear sky conditions, EPIC-based total SWs capture the diurnal variation well with small RRMSEs ranging from 10.10%~16.24%. For cloudy sky, EPIC-based total SWs have better performance with RRMSEs smaller than 10.02%. For overcast sky conditions, EPIC-based products overestimate the total SWs with RRMSEs larger than 29.20%. It is noteworthy that EPIC-derived products show the worst performance for SURFRAD-BOS sites, likely caused by the rugged terrain around this site. For diffuse SWs shown in **Fig. S1**, clear-sky EPIC-derived estimates have the largest RRMSEs. For direct

SWs, **Fig. S2** shows that overcast-sky EPIC-based estimates have large uncertainties due to their relatively small magnitude. **Fig. S3** shows that EPIC-based total PARs perform better than total SWs, especially for clear and cloudy sky conditions.

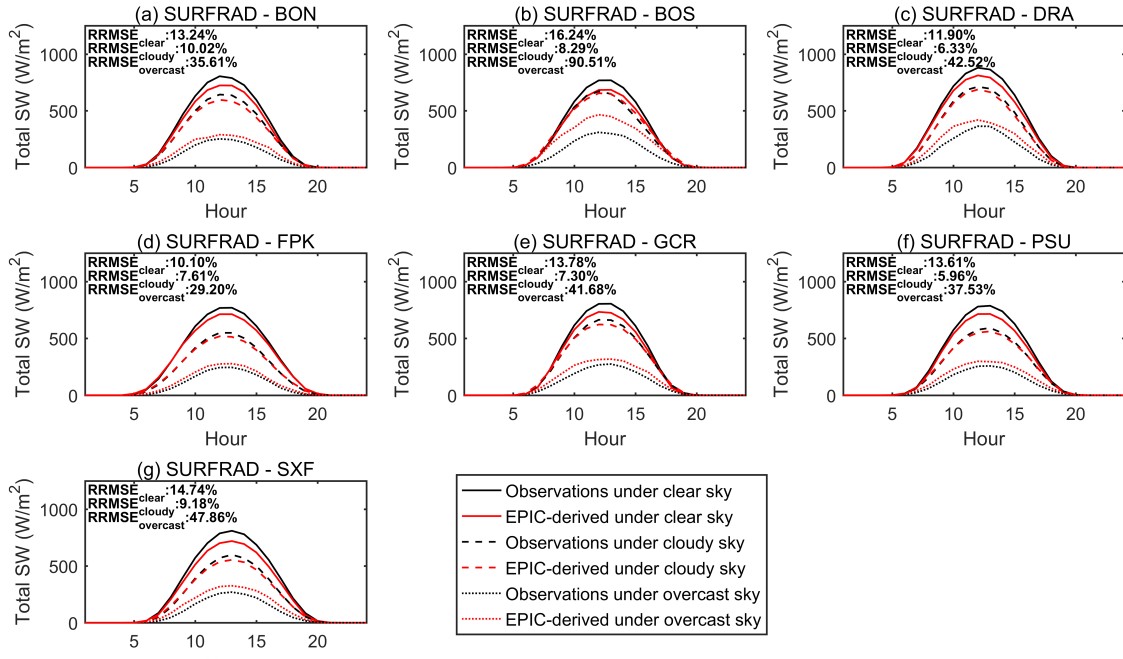

**Figure 4.** Diurnal variations of both EPIC- and ground-based total SWs, averaged, from June, 2015 to June, 2019 for different sky conditions at 7 SURFRAD sites.


We further analysed the accuracy of our products at both different local times and for different months. **Fig.5** shows that the accuracy of hourly SW/PAR estimates changes with the change in local time. The SW/PAR estimates for nearly local noon have negative biases, larger RMSEs but smaller RRMSEs, where those for early morning or later afternoon have positive biases, smaller RMSEs but larger RRMSEs. However, $R^2$ values of total SW and PAR estimates are generally larger than 0.7 for all local times.

The daily SW/PAR estimates show good accuracy for all months. Total SW and PAR estimates from May to August have positive biases, larger RMSEs but smaller RRMSEs. The $R^2$ values of SW/PAR estimates show little monthly and seasonal dependency. These results confirm that both EPIC-based hourly and daily products have reliable accuracy, although the magnitudes of SW/PAR, and cloud and atmospheric conditions at different time (i.e. hour, day, and month) may affect the accuracy and uncertainties of these products.

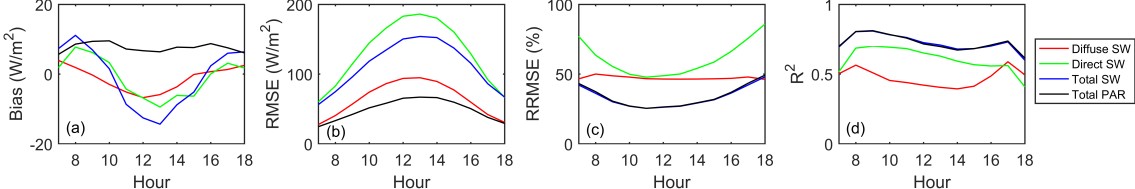


**Figure 5.** Evaluation of EPIC-based hourly SW/PAR estimates at different local hours from 7:00 to 18:00.



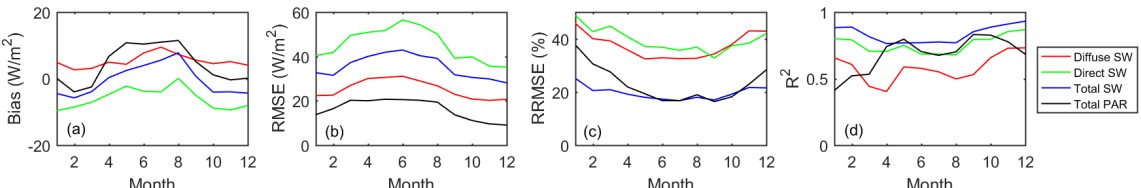

**Figure 6.** Evaluation of EPIC-based daily SW/PAR estimates on different months during the study period.

### 3.1.3 Spatial distribution of estimated errors of derived products

The hourly and daily total SW estimates show similar spatial patterns in their accuracy statistics composed of biases, RMSEs, RRMSEs and $R^2$ values (**Figs. 7 and 8**). Polar regions have relatively small RMSE but large RRMSE, due to long-term or frequent ice/snow cover in these regions and a lack of proper accounting for land surface albedo in current products; the island and coastal regions show the worst performance with high bias, large RMSE and low $R^2$; and derived products for most of the continental sites perform well but show large spatial heterogeneities related to different land cover types, climate zones, surface topography, etc.

The BSRN-IZA site, a high-mountain station located in Tenerife (Canary Islands, Spain), exhibit high negative bias and large errors and uncertainties, which can be explained by its geographic location in the Tenerife island within Teide volcano area (García et al., 2019), and the particular weather conditions for this area where the clouds affect the lower parts of the island (below 2000 m above sea level) while the sky for upper parts probably remains clear (Urraca et al., 2018). In general, EPIC-derived products have higher accuracy in continental regions with low bias and small RMSE and RRMSE, whereas island or coastal regions show very

large bias, and large RMSE and RRMSE values (**Tables 3 and 4)**, probably caused by the rapidly changing weather condition and the mixture of land and water in a grid cell (edge effects), which is identical with other studies (Boland et al., 2016;Wang and Pinker, 2009)

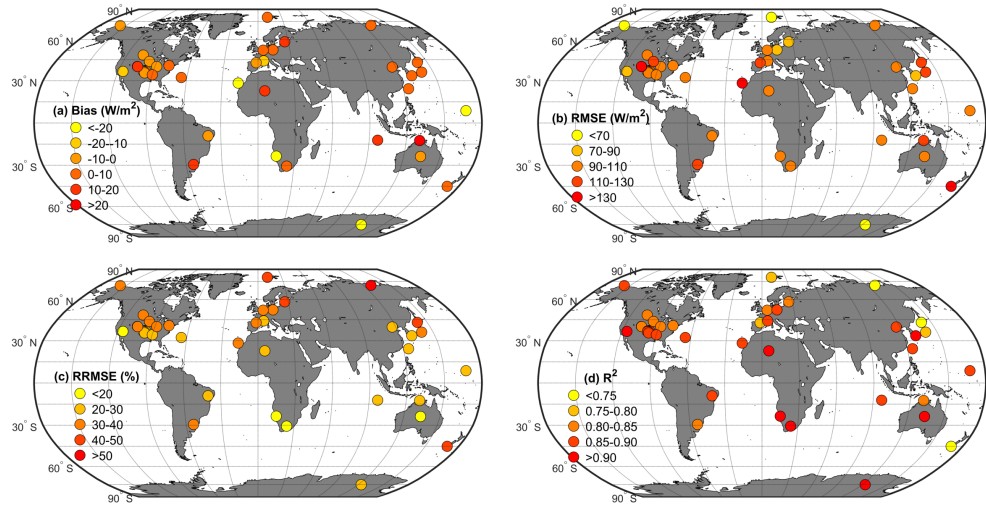

**Figure 7.** Spatial distributions of accuracy statistical metrics for EPIC-based hourly total SW at all ground-based sites. Circles
with different colors indicate their different values.

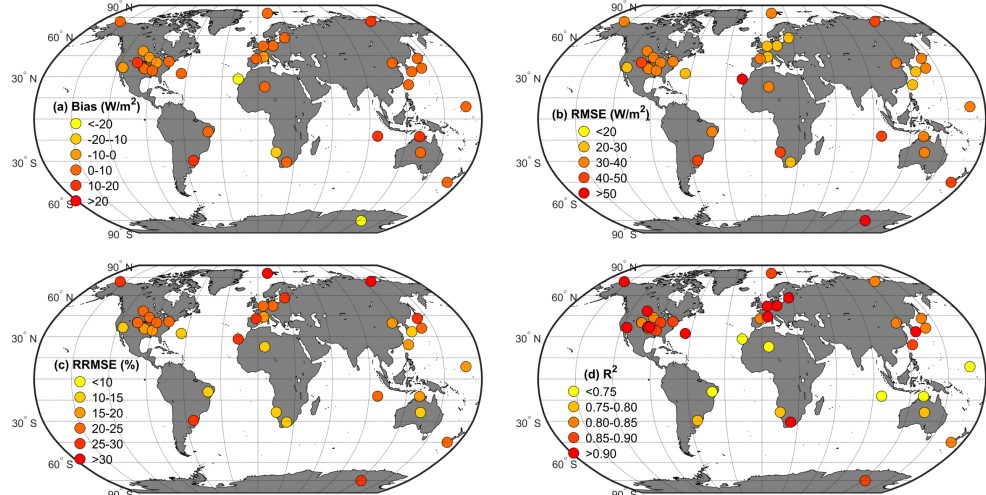

**Figure 8.** Spatial distributions of accuracy statistical metrics for EPIC-based daily total SW at all ground-based sites. Circles with different colors indicate their different values.


**Table 1.** Evaluation of EPIC-based hourly SW estimates against the ground-based measurements for different regions.

| Region | Parameter | Bias (W/m²) | RMSE (W/m²) | RRMSE (%) | R² |
|---|---|---|---|---|---|
| Polar | Diffuse SW | 4.11 | 60.22 | 53.78 | 0.68 |
| | Direct SW | -11.70 | 85.37 | 104.54 | 0.63 |
| | Total SW | -3.12 | 65.69 | 37.34 | 0.85 |
| Island/Coastal | Diffuse SW | 66.65 | 125.57 | 78.76 | 0.54 |
| | Direct SW | -109.11 | 224.06 | 68.26 | 0.58 |
| | Total SW | -32.25 | 134.13 | 28.90 | 0.83 |
| Continental | Diffuse SW | 4.43 | 69.05 | 50.80 | 0.62 |
| | Direct SW | -9.27 | 130.53 | 50.14 | 0.76 |
| | Total SW | -1.07 | 102.97 | 27.59 | 0.87 |

**Table 2.** Evaluation of EPIC-based daily SW estimates against the ground-based measurements for different regions.

| Region | Parameter | Bias (W/m²) | RMSE (W/m²) | RRMSE (%) | R² |
|---|---|---|---|---|---|
| Polar | Diffuse SW | 2.54 | 26.79 | 37.45 | 0.82 |
| | Direct SW | -2.57 | 40.28 | 90.98 | 0.71 |
| | Total SW | 0.28 | 35.96 | 31.65 | 0.88 |
| Island/Coastal | Diffuse SW | 34.58 | 48.33 | 61.66 | 0.45 |
| | Direct SW | -46.72 | 84.87 | 56.68 | 0.50 |
| | Total SW | -12.07 | 53.94 | 23.65 | 0.67 |
| Continental | Diffuse SW | 2.39 | 21.27 | 31.91 | 0.69 |
| | Direct SW | -2.05 | 39.45 | 33.70 | 0.82 |
| | Total SW | 0.40 | 32.80 | 17.94 | 0.88 |



## 3.2 Globally spatio-temporal patterns of derived products

We investigated the spatial patterns of averaged total SWs during the three whole years of 2016-2018 for different seasons : 1) Spring, March, April, and May (MAM); 2) Summer, June, July, and August (JJA); 3) Autumn, September, October, and November, (SON); 4) Winter, December, January, and February (DJF). **Fig. 9(a-d)** show the EPIC-based products reflect the heterogenous spatial distributions and track the globally seasonal variations that are mainly due to the sun angle variations. They also have a consistent pattern when compared with the CERES-derived products (**Fig. 9 (e-f)**). **Fig. S4** shows that EPIC- and CERES-derived estimates have small differences over most regions, especially in Spring, Autumn and Winter, but some large discrepancies are observed in the Tibetan Plateau due to the frequent ice/snow cover and in the Congo basin due to the complex cloud and atmospheric conditions. However, our EPIC-derived products can reveal more spatial details than CERES-derived estimates due to their higher spatial resolution.

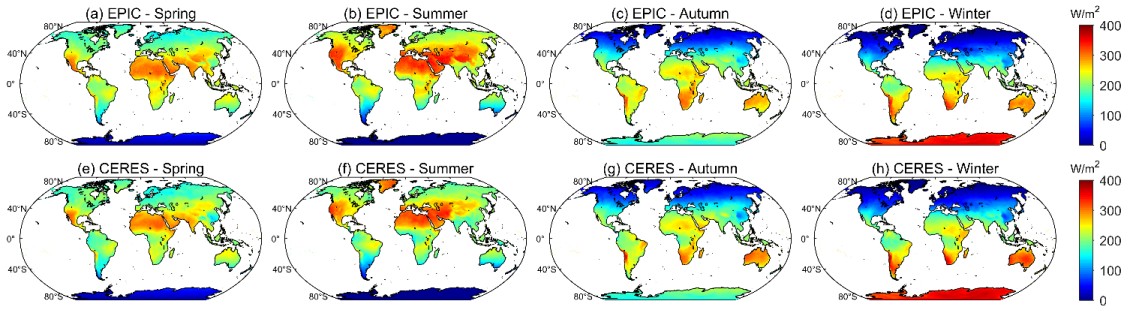

**Figure 9.** Global distributions of EPIC- and CERES-derived total SW fluxes for different seasons during the study period, 2016-2018.

**Fig. 10** shows the temporal variations of both EPIC- and CERES-based monthly total SWs for the land surface of globe, northern and southern hemispheres during June, 2015 to June, 2019. All EPIC- and CERES-based products show similar temporal variations. From Autumn (SON) to next Spring (MAM), EPIC-based global SWs coincide well with CERES-derived ones, whereas in Summer (JJA), EPIC-based global SWs are lower than CERES-derived ones, due to the differences in northern hemisphere. The differences in spatial resolution, driving data, retrieval models/algorithms, etc. contribute to these discrepancies. **Fig. 11** displays the temporal variations of daily zonal averaged total SWs and PAR products. EPIC- and CERES-based estimates present highly consistent latitude-gradient distributions and temporal variations. **Fig. S5** shows the differences between EPIC- and CERES-based SW/PAR estimates. Overall, total SW and PAR and their direct and diffuse components have small differences, but the direct and diffuse components of SW show relatively large differences in the northern hemisphere, possibly due to the underestimation of CERES-based direct components and overestimation of CERES-derived diffuse components (Hao et al., 2019).
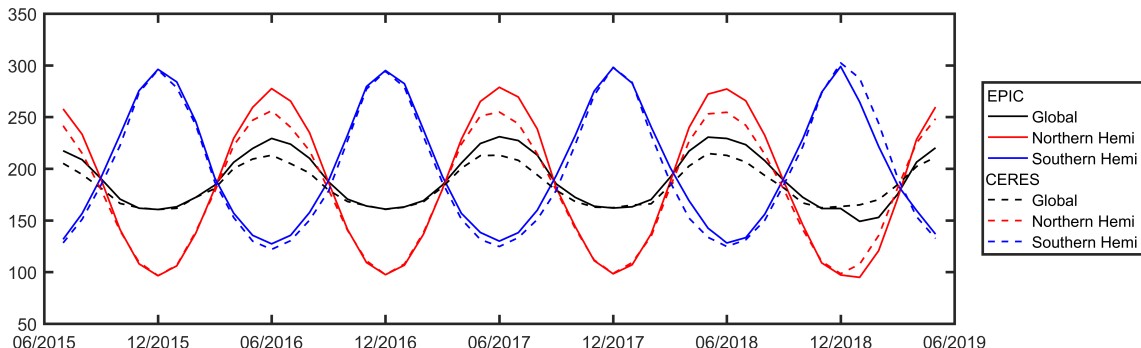

**Figure 10**. Temporal variations of EPIC- and CERES-based monthly total SW for the land surface of global, northern- and southern-hemispheres.

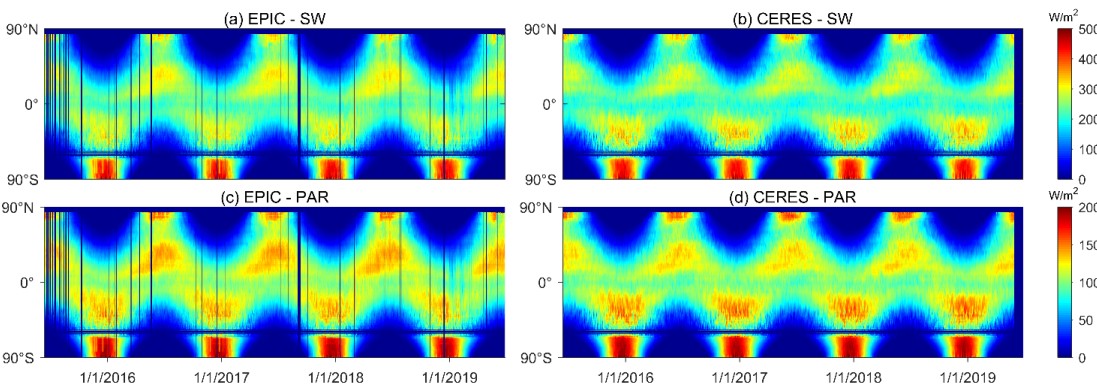

**Figure 11.** Temporal variations of EPIC- and CERES-based zonally-averaged daily total SW and PAR for global land areas. The vertical dark bars in **(a, c)** are due to the missing data.

## 4 Discussion

The proposed new SW/PAR products presented here make full use of the unique advantages of DSCOVR/EPIC, an instrument that observes nearly the entire sunlit areas of the Earth, from pole to pole, every 1~2 hours. These promising products have some

unique attributes: 1) show high correlations with the ground station observations; 2) present reasonable and identical spatio-temporal SW/PAR patterns but more spatial details when compared to CERES products; and 3) capture accurately the diurnal cycles of SW/PAR. In addition, they are based on a single instrument, EPIC, and thus avoid the sensor-to-sensor differences and inconsistencies inherent in multi-source datasets. The DSCOVR/EPIC science team is continuing to refine their algorithms and upgrade the product quality, and we plan to update our products accordingly.

We expect and hope these products will have multiple uses in diverse earth science communities. When combined with the DSCOVR/EPIC-derived vegetation data, our globally high-accuracy products can be used to understand the diurnal variabilities and underlying processes of photosynthesis and evapotranspiration for terrestrial ecosystems. By virtue of the decomposition of the direct and diffuse PARs, our products are expected to improve the estimates of ecosystem photosynthesis capacity and primary production. Our products may also be valuable for site selection for solar power production and solar energy management.

Some sources of uncertainties probably affect the accuracy and reliability of our products (see **Figs 2 and 3**). There are some geolocation, inter-calibration and misregistration issues in the current EPIC L1B version 2 products (Molina García et al., 2019),



which will be improved in the version 3 products (announced in the DSCOVR 2019 Science Team Meeting, Greenbelt, MD) in future. The current EPIC L2 atmospheric and cloud products have large uncertainties or gaps in the ice/snow covered regions and when the solar angle is large (>70°) (Yang et al., 2019;Herman et al., 2018;Xu et al., 2017). We used the CERES data to fill the

gaps, especially in the polar regions. To allow users maximum flexibility, we also provided quality flags that indicate whether pixels are successfully retrieved or gap-filled in the derived products. Current algorithms/models neglect the impact of water vapor and land surface albedo, which could lead to some additional biases and uncertainties. This problem will be addressed through the combination with high-quality EPIC-view water vapor products such as the EPIC-view Multi-Sensor Global Cloud and Radiance Composites (Khlopenkov et al., 2017) and the development of land surface albedo products based on the EPIC Multi-Angle

Implementation of Atmospheric Correction (MAIAC) products (Hao et al., 2019). We also did not account for the impacts of spatial mismatch between ground-based observation and EPIC-based data. In the future, we will collect long-term, high-quality and widely-distributed ground-based datasets to improve and evaluate our products.

The spatial resolution of our current products is relatively coarse (0.1°×0.1°). The effective spatial resolution of the original EPIC image is relevant to the observing angle, and is about 10km at nadir (near the centre of the image) and 20km at 60° (Marshak

et al., 2018). Higher spatial resolution SW/PAR products are desired for mapping of carbon and water fluxes as well as solar energy assessment and operation, especially for islands and coastal regions (see **Tables 3 and 4**). Spatial scale mismatch between land surface properties and derived SW/PAR estimates can limit the applications of our products (Ryu et al., 2018). We suggest that spatial downscaling techniques can be used to improve our proposed products (Wang et al., 2019), especially for rugged terrain (e.g. the Tibetan Plateau), where topographic effects (e.g. varied elevation, rotation of solar geometry, shadowing, terrain occlusion, and multi-scattering) on SW/PAR should be considered and assessed (Zhang et al., 2019;Hao et al., 2018b;Hao et al., 2018a). We

believe such analysis/assessment can benefit greatly next generation mission of deep-space satellites/sensors such as DSCOVR/EPIC, for optimizing/balancing the trade-off between data amount (spatial/temporal resolution) and transmission time, perhaps leading to improved spatio-temporal resolution of future data products from such missions.

Finally, one shortcoming of current products is the relatively short period of the derived data records which cover only 4 years.

Such record length is not adequate to detect any globally long-term trends. A feasible solution is to merge DSCOVR/EPIC products with reanalysis data/products to produce globally continuous, consistent and long-term SW/PAR datasets, through correcting the reanalysis data based on satellite data (Feng and Wang, 2018). Although currently DSCOVR is temporarily in safe mode, it is expected to return to operations early in 2020 (https://epic.gsfc.nasa.gov/). With the increasing record length of EPIC data, it is anticipated that the temporal coverage of our proposed products will be also extended to support the long-term analysis in the

future.

## 5 Data availability

Both the derived hourly and daily DSCOVR/EPIC-based global SW/PAR products are available at the DataHub (https://doi.org/10.25584/1595069, Hao et al., 2020), Pacific Northwest National Laboratory (PNNL). The hourly data are grouped by day in distinct NetCDF files, which are named as "EPIC_SW_PAR_Hourly_yyyymmdd.nc" where "yyyy", "mm", and "dd"

denote year, month, and day (UTC time). The daily data are grouped by month in distinct NetCDF files, which are named as "EPIC_SW_PAR_Daily_yyyymm.nc" where "yyyy", and "mm" denote year and month (UTC time). Each NetCDF file contains latitude, longitude, time, diffuse SW, direct SW, diffuse PAR, direct PAR, and the corresponding quality flags which indicate whether the pixel is gap-filled or not. The scale factor for the direct and diffuse SW/PAR is 0.1. The total SW/PAR estimates can

be calculated by combining the direct and diffuse components. The information about the version, creation date, reference, contact

mails, and other comments are also included in the file.

## 6 Conclusions

This paper presents the first globally hourly and daily SW/PAR products, with a spatial resolution of 0.1°×0.1° for the period of June, 2015~June, 2019 based on the DSCOVR/EPIC observations. The newly developed products are the first of their kind because of high temporal frequency (hourly) and global coverage at a spatial resolution of 0.1°×0.1°, only based on a single instrument,

DSCOVR/EPIC. We evaluated the EPIC-derived products against the globally-distributed ground-based data, and analysed and compared the spatio-temporal variations of the proposed products with the well-characterized and widely-used CERES products. EPIC-derived SW/PAR estimates and their direct and diffuse components show good consistencies with the globally-distributed ground-based observations. The EPIC-derived products capture accurately the diurnal variabilities of SW/PAR under different sky conditions. The comparisons with CERES data indicate that the developed products reflect complex spatial heterogeneities and

capture substantial seasonal variabilities of SW/PAR effectively with the same temporal resolution of hourly but higher spatial resolution. The promising products offer an invaluable resource for solar photovoltaic applications and understanding and exploring the diurnal cycles of terrestrial water, carbon, and energy fluxes at various temporal and spatial scales. We plan to update our proposed products as additional EPIC observations become available, and with further improvements of the record length of EPIC data and algorithm refinements that are planned by the EPIC/DSCVR science team in the future.

**Author contributions**

CM and HD designed the study. HD and CM produced the products, analyzed the results, and drafted the original paper. All authors contributed to the analysis and interpretation of the results, and to improving this paper.

**Competing interests**

The authors declare that they have no conflict of interest.

**Acknowledgements**

The CERES and CAVE data were obtained from the NASA Langley Research Center Atmospheric Science Data Center. We thank the entire DSCOVR/EPIC team for the powerful L2 products and especially Dr. Pam Mlynczak for valuable information on CERES Edition 4.1 products. We also thank Dr. Ben Bond-Lamberty for the valuable suggestions, and Emmanuel Bonilla for his help on sharing the data on the DataHub of PNNL.

**Financial support**

This work was supported by a Laboratory Directed Research and Development project sponsored by the Pacific Northwest National Laboratory (PNNL) of the U.S. Department of Energy. Dalei Hao was sponsored by China Scholarship Council. The research was performed using resources available through Research Computing at PNNL. PNNL is operated by Battelle for the U.S. Department of Energy under Contract DE-AC05-76RL01830.





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
