# Peer review of "DSCOVR/EPIC-derived global hourly/daily downward shortwave and photosynthetically active radiation data at 0.1°×0.1° resolution"

_Earth System Science Data, 2020_

## Referee Comment (RC1) · Anonymous Referee #1 · 16 May 2020

General comments: This authors introduced their newly developed global surface SW and PAR products derived based upon DSCOVR-EPIC observations. A detailed evaluation is given by comparing the products with surface-based observations. The paper is organized and presented very well. The dataset is an advancement to the field and has the potential for more applications. From the Earth's L1 point, EPIC's view of the entire sunlit side of the Earth makes it very suitable for this work and the authors developed a convincing way to retrieve the SW and PAR info. I recommend publication of the paper after some minor revisions listed below.

Specific comments: 1) P3, Line80: "DSCOVR/EPIC has the potential to reduce signifi-

[Figure]

cantly or eliminate completely the 3D radiative 80 effects on the final products". This is a good point, but I suggest more discussion and define the 3D effects you are referring to. 3D effect has different meaning for different discipline. For example, for cloud or surface property retrievals, one 3D effect pathway is that photons can come to the field of view when scattered by the surrounding clouds. This could happen regardless of the sun-view geometry. I see your point here and I agree that it is a big advantage of EPIC for this application. More elaboration will make this more clear. 2) P3, Line106: please check "gridded monthly, monthly hourly, daily," is "monthly hourly" a typo? 3) P3, Line108: "However, the SYN1deg Edition 4.1 products are not suitable for inferring long-term trends of surface fluxes, due to limited climate quality". Need reference or more elaboration. 4) Figure 10: vertical axis label missing 5) P13, Line307-308: "Although currently DSCOVR is temporarily in safe mode, it is expected to return to operations early in 2020 (https://epic.gsfc.nasa.gov/)." DSCOVR is back and operational since Feb. 2020.

---

## Referee Comment (RC2) · Meredith Brown (Referee) · 8 Jul 2020

The authors present a unique use of EPIC data for SW/PAR retrievals. Potentially useful to the global climate modeling community and surface radiation budget studies.

The spatial resolution may be too coarse for many vegetation studies, but the solar power communities may find this very useful, especially the temporal resolution.

Overall, a fine paper, it is well structured and clearly written.

---

## Author Comment (AC1) · 15 Jul 2020

General comments: This authors introduced their newly developed global surface SW and PAR products derived based upon DSCOVR-EPIC observations. A detailed evaluation is given by comparing the products with surface-based observations. The paper is organized and presented very well. The dataset is an advancement to the field and has the potential for more applications. From the Earth's L1 point, EPIC's view of the entire sunlit side of the Earth makes it very suitable for this work and the authors developed a convincing way to retrieve the SW and PAR info. I recommend publication of the paper after some minor revisions listed below.

Thank you for your positive reviewing comments.

Specific comments:

P3, Line80: "DSCOVR/EPIC has the potential to reduce significantly or eliminate completely the 3D radiative 80 effects on the final products". This is a good point, but I suggest more discussion and define the 3D effects you are referring to. 3D effect has different meaning for different discipline. For example, for cloud or surface property retrievals, one 3D effect pathway is that photons can come to the field of view when scattered by the surrounding clouds. This could happen regardless of the sun-view geometry. I see your point here and I agree that it is a big advantage of EPIC for this application. More elaboration will make this more clear.

Thank you for this excellent suggestion. In the revised manuscript, we first added a definition of the 3D effects of cloud including the nonlocal cloud shadows, reflections from cloud sides, and enhancement of downward radiation by photon diffusion from clouds (Line 63-64). We further clarified that DSCOVR/EPIC can reduce significantly the 3D radiative effects caused by the shift of the apparent position of clouds and their shadows which are related to the solar and viewing geometries (Line 81-82).

P3, Line106: please check "gridded monthly, monthly hourly, daily," is "monthly hourly" a typo?

Thank you for pointing this typo error. We deleted 'monthly hourly' in the revised manuscript.

P3, Line108: "However, the SYN1deg Edition 4.1 products are not suitable for inferring long-term trends of surface fluxes, due to limited climate quality". Need reference or more elaboration.

This was reported by the Data Quality Summary published by the CERES team in https://ceres.larc.nasa.gov/documents/DQ_summaries/CERES_SYN1deg_Ed4A_DQS.pdf. We added the reference in the revised manuscript.

Figure 10: vertical axis label missing

We added the y label "SW (W/m2)" to the revised figure 10.

P13, Line307-308: "Although currently DSCOVR is temporarily in safe mode, it is expected to return to operations early in 2020 (https://epic.gsfc.nasa.gov/)." DSCOVR is back and operational since Feb. 2020.

Yes. Thank you. We have deleted this statement in the revised manuscript.

---

## Author Comment (AC2) · 15 Jul 2020

The authors present a unique use of EPIC data for SW/PAR retrievals. Potentially useful to the global climate modeling community and surface radiation budget studies. The spatial resolution may be too coarse for many vegetation studies, but the solar power communities may find this very useful, especially the temporal resolution. Overall, a fine paper, it is well structured and clearly written.

Thank you for your positive comment!

––––––––––––––––––––––

2020.